# Flocculant-Assisted Synthesis of Graphene-Like Carbon Nanosheets for Oxygen Reduction Reaction and Supercapacitor

**DOI:** 10.3390/nano9081135

**Published:** 2019-08-07

**Authors:** Yinglin Zhang, Yulin Shi, Bo Yan, Tingting Wei, Yin Lv, Long Chen, Feng Yu, Xuhong Guo

**Affiliations:** 1Key Laboratory for Green Processing of Chemical Engineering of Xinjiang Bingtuan, School of Chemistry and Chemical Engineering, Shihezi University, Shihezi 832003, China; 2State Key Laboratory of Chemical Engineering, East China University of Science and Technology, Shanghai 200237, China

**Keywords:** chitosan, textile sludge, graphene-like carbon nanosheets, oxygen reduction reaction, supercapacitor

## Abstract

The rational treatment of hazardous textile sludge is critical and challenging for the environment and a sustainable future. Here, a water-soluble chitosan derivative was synthesized and used as an effective flocculant in removal of reactive dye from aqueous solution. Employing these chitosan-containing textile sludges as precursors, graphene-like carbon nanosheets were synthesized through simple one-step carbonization with the use of Fe (III) salt as graphitization catalyst. It was found that the resultant graphene-like carbon nanosheets material at thickness near 3.2 nm (NSC-Fe-2) showed a high graphitization degree, high specific surface area, and excellent bifunctional electrochemical performance. As-prepared NSC-Fe-2 catalyst exhibited excellent oxygen reduction reaction (ORR) activity (onset potential 1.05 V) and a much better methanol tolerance than that of commercial Pt/C (onset potential 0.98 V) in an alkaline medium. Additionally, as electrode materials for supercapacitors, NSC-Fe-2 also displayed an outstanding specific capacitance of 195 F g^−1^ at 1 A g^−1^ and superior cycling stability (loss of 3.4% after 2500 cycles). The good electrochemical properties of the as-prepared NSC-Fe materials could be attributed to the ultrathin graphene-like nanosheets structure and synergistic effects from codoping of iron and nitrogen. This work develops a simple but effective strategy for direct conversion of textile sewage sludge to value-added graphene-like carbon, which is considered as a promising alternative to fulfill the requirements of environment and energy.

## 1. Introduction

Reactive dyestuffs have attracted a great deal of interest for textile dyeing industry in recent years due to its highly saturated color, good wet fastness, and brightness [1,2]. Total world consumption of reactive dyes is estimated as 80,000 tons, ~50–60% of which is lost and cannot be reused during the dyeing process [1,3]. Moreover, in the reactive dyeing process, a large amount of salt (up to 150 g L^−1^) is often used to promote dyeability between the negatively charged cotton and the anionic dyes [4,5]. As a result, some traditional wastewater treatment processes, such as biological methods, might become ineffective for the treatment of high salinity dyeing effluent [6]. Coagulation–flocculation is a primary method for treating textile dyeing wastewater due to its low-cost and high efficiency [7]. However, it generates large amounts sludge, which takes further controversy on the disposal of hazardous waste properly.

Landfill and incineration have been widely used for the treatment of sewage sludge due to their economic advantages [8]. However, textile sludge is characterized by their fairly deep color and high content of heteroatom [9]. Therefore, the lower quality landfilling of dye sludge at wastewater treatment plant may result in groundwater contamination due to generation of heavily polluted leachates. The major potential environmental impacts related to incineration are pollution of air from sulfur (S) and nitrogen oxides (NO*_x_*) emissions [10]. Hence, there is a need to manage this waste in an ecologically sustainable way.

Pyrolysis of sludge to produce carbonaceous materials is a promising method for the reuse and recycle of sewage sludge. The advantage of using sewage sludges as raw materials for manufacturing carbon-based materials is that these waste sludge have higher organic content and the very low cost [11]. Besides its economic benefits, pyrolytic conversion process renders hazardous waste to decompose into less hazardous or nonhazardous material under controlled conditions. Because the activated carbon has been known as perfect adsorption ability for organic compounds, many researchers are paying attention to synthesizing activated carbons from textile sewage sludge by physiochemical activation [12]. These sludge-derived porous carbon materials are widely used in environmental management as a low adsorbent [13]. In general, the total operating costs of wastewater treatment will involve the sewage sludge disposal cost. Therefore, conversion of sludge to a value-added product has been considered as a possible compensation measure to reduction of overall wastewater treatment cost [14]. The recycling of textile sewage to produce advanced electrode material for energy applications is of highly practical and attractive prospect.

Two-dimensional (2D) materials are layered crystalline materials which built from of a single layer nanosheet on a 2D honeycomb lattice [15]. Among various 2D materials, graphene-like carbon has attracted much interest due to their potentials in next-generation energy conversion and storage devices [16]. However, the lack of cost-effective raw materials and ton-level preparation of graphene-like carbon limits their further applications in the market. Taking full advantage of the film-forming property of polymer, Primo et al. and coworkers synthesized graphene-like N-doped porous carbon materials by using chitosan as a raw material [17]. More recently, Hao et al. developed a simple method to prepare graphene-based carbon aerogels employing chitosan as a starting materials [18]. On the other hand, a number of studies have demonstrated that chitosan, a natural cationic polymer, can be used as an effective flocculant to purify wastewater, particularly with respect to textile wastewater containing anionic reactive dyes [19]. Utilizing this chitosan-containing textile sludge as precursors, graphene-like carbon nanosheets are more likely to be produce on a large scale and at low cost.

In this work, a water-soluble chitosan derivative was synthesized and used as an effective flocculant in removal of reactive dye from aqueous solution. The possible composition and structure of sewage sludge was investigated by stoichiometric charge neutralization. Subsequently, graphene-like carbon nanosheets material were obtained by pyrolysis of chitosan-containing textile sludge with the use of Fe (III) salt as graphitization catalyst. The resultant graphene-like carbon nanosheets material (NSC-Fe-2) showed a high graphitization degree, high specific surface area, and excellent bifunctional electrochemical performance. NSC-Fe-2 as catalyst showed excellent ORR activity and a much better methanol tolerance than that of commercial 20 wt.% Pt/C. In addition, NSC-Fe-2 as electrode material also displayed much superior specific capacitance (195 F g^−1^ at 1 A g^−1^) and good cycling stability for supercapacitor application. The strategy of textile sludge conversion to value-added graphene-like carbon is considered as a promising alternative to fulfill the requirements of environment and energy.

## 2. Experimental

### 2.1. Chemicals

A commercially azo dye, reactive brilliant red (K-2BP; M*_w_* = 808.5 g mol^−1^; λ_max_ = 537.4 nm, structure is shown in Appendix A), was obtained from Longsheng Dyestuff Chemical Co., Ltd. (Shandong, China). Cationic flocculant 2,4-bis(dimethyl amino)-6-chloro-[1,3,5]-triazine-chitosan (BDAT-CTS) was synthesized based on previous reports, the structure is shown in Appendix A [20]. Chitosan (85% deacetylated; M*_w_* = 2.5 × 10^5^ g mol^−1^), ferric sulfate (analytical grade), potassium hydroxide (KOH, 95%), nickel foam, Nafion^®^ perfluorinated resin solution (5 wt.%), acetylene black, commercial Pt/C (20 wt.%), and a polytetrafluoroethylene (PTFE) dispersion (60 wt.%) were obtained from Aladdin Reagent Co., Ltd. (Shanghai, China). Distilled H_2_O was used in all experiments.

### 2.2. Preparation of Chitosan-Containing Textile Sludge

To simulate actual reactive dye wastewater, the K-2BP must be hydrolyzed before use. The route of K-2BP hydrolysis is shown in Appendix A. All flocculation experiments were performed at room temperature using a series of 100-mL beakers. The flocculant solution (1 g L^−1^) was obtained via dissolving the flocculant (BDAT-CTS). The dye wastewater solution (0.2 g L^−1^) was obtained via dissolving K-2BP. Specifically, 0.2–5 mL of flocculant solution was added to 30 mL of the dye liquor, and then water was added to reach an 80 mL volume. The pH of the solution was adjusted to 1.6 by 1 M HCl after stirring the above solution; thereafter, the solution was allowed to precipitate for 1 h. At this time, the concentration of the dye liquor was 75 mg·L^−1^. After dye flocculation, the zeta potential was tested. Then, the filtrate was collected and evaluated by UV–Vis spectrophotometry at λ_max_.

Dye removal (*R*%, Equation (1)) and flocculation capacity (*Q*, Equation (2)) were calculated as follows
(1)$$R\%=\frac{C_{0}-C_{i}}{C_{0}}\times 100$$
(2)$$Q=V\frac{C_{0}-C_{i}}{W}$$
where *C*_0_ and *C_i_* (mg·L^−1^) are the initial dye concentration and the concentration after flocculation, respectively; *V* (mL) is the solution volume and *W* (g) is the mass of the BDAT-CTS.

### 2.3. Preparation of NSC-Fe

The preparation process of NSC-Fe-*x* is illustrated in Scheme 1. The chitosan-containing textile sludges were dried at 80 °C and then ground to powder for later use. Typically, the 0.06 g Fe_2_(SO_4_)_3_ was first mixed with 0.3 g textile sludge in an agate mortar. The crushed powder was then placed in a combustion boat and heated to 800 °C in a quartz tubular reactor with a 50 mL min^−1^ argon flow for 2 h. Finally, the products were washed with 2 M HCl (20 mL) to free inorganic matter, washed with water to neutral, and finally dried at 80 °C for 10 h. The as-prepared product was labeled NSC-Fe-*x* (*x* = 1, 2, and 3), corresponding to the initial ratio between the Fe_2_(SO_4_)_3_ and chitosan-containing textile sludge, which were 0.1, 0.2, and 1, respectively.

### 2.4. Characterization

X-ray diffraction (XRD) patterns were recorded on a Bruker D8 ADVANCE (Zeiss, Karlsruhe, Germany) apparatus with Cu-Kα radiation. Raman spectra were obtained using a Raman spectrometer (Bruker Senterra, Renishaw Invia, Renishaw, Wotton-under-Edge, Gloucestershire, UK) under a laser excitation of 532 nm. The N_2_ adsorption/desorption isotherms were determined at 77 K using a MicrotracBEL BELSORPmax apparatus (JWGB, Beijing, China). The surface morphologies were determined by scanning electron microscopy (SEM, Nova NanoSEM 200, Zeiss, Karlsruhe, Germany). The detailed structures of the samples were examined via transmission electron microscopy (TEM), high-resolution TEM (HRTEM), and element mapping (JEM-2100, working at 200 kV, JEOL, Tokyo, Japan). The thickness of NSC-Fe was acquired by atom force microscopy (AFM; Multimode Nanoscope 8, Karlsruhe, Baden-Wurttemberg, Germany). X-ray photoelectron spectroscopy (XPS) was recorded on an ESCALAB 250Xi (Thermo Fisher Scientific Company, Waltham, MA, USA) apparatus.

### 2.5. Electrochemical Measurements

A Chenhua CHI 760D electrochemical workstation (Shanghai, China) was used to evaluate the supercapacitance and oxygen reduction reaction (ORR) electrochemical properties.

Electrocatalytic ORR activity: The catalyst ink was prepared by ultrasonically mixing Nafion^®^ perfluorinated resin solution (25 µL) and catalyst (3 mg) for 20 min in 475 µL of ethanol. Afterwards, the obtained catalyst ink (10 µL) was deposited onto a glassy carbon electrode (GCE, 3 mm in diameter) and used as the substrate for the working electrode (WE). Pt foil was used as the counter electrode (CE) and Ag/AgCl (3.5 M KCl solution) as the reference electrode (RE). The electrolyte used here was solution of 0.1 M KOH. The potential with a current density of 0.1 mA cm^−2^ to the electrochemical test section was used as the onset potential. All potentials are calibrated for reversible hydrogen electrodes (RHE).

Supercapacitors: Cyclic voltammetry (CV) and galvanostatic charge/discharge (GCD) measurements were used to estimate the capacitive properties of the electrode materials. Overall tests were conducted in a three-electrode device (6 M KOH). The WE was prepared via ultrasonically mixing PTFE (1 µL), carbon material (5 mg), and acetylene black (1 mg) for 30 min in 1 mL of ethanol. The prepared ink was dropped on nickel foam (1 cm × 1 cm) and dried at 80 °C for 12 h. Then, the nickel foam was pressed at 25 MPa for 20 s and was activated in 6 M KOH for 6 h. For the three-electrode system, the CE was a Pt slice, and a saturated calomel electrode (SCE) served as the RE.

The GCD specific capacitance was calculated as follows
(3)$$C=\frac{I\Delta t}{m\Delta V}$$
where *C* (F g^−1^) is the specific capacitance, Δ*t* (s) is the discharge time, *I* (A) is the discharge current, Δ*V* (v) is the voltage range, and *m* (g) is the mass of the working electrode.

## 3. Results and Discussion

### 3.1. Flocculation Performance

The flocculant concentration was a significant controlling factor in the flocculation system. The effect of the flocculant concentration on the removal rate and the zeta potential (ZP) is displayed in Figure 1a. The removal rate first increased and then decreased with increasing flocculant concentration (from 2.5 to 62.5 mg·L^−1^). When the flocculant concentration was low, the neutralization impact was feeble, leading to a low removal rate. The optimum removal rate (*R* > 99.2%) was achieved when the flocculant dose was 32.5 mg·L^−1^, and the ZP of the solution approached zero at that concentration, indicating that the negative charges of the dye were completely neutralized by the flocculant, resulting in a high removal rate. When the optimum concentration was exceeded, the existence of superfluous protonated amine groups on the flocculant led to restabilization of the suspension [21], which caused a lower removal rate and higher ZP. The effects of the flocculant concentration on the flocculation capacity are shown in Figure 1b. When the flocculant concentration was 32.5 mg·L^−1^, the flocculation amount reached 2.29 g g^−1^.

When the dye was dissolved in water, the sulfonate group (D-SO_3_Na) of the dye dissociated, and the dye was transformed into an anionic dye. Moreover, the amino group of the flocculant was protonated under strong acid conditions, and then the flocculation was carried out by electrostatic attraction. The flocculation factor n (n: mol dye/mol amine groups) can be calculated according to the average nitrogen content of the flocculant [20]. At the optimum flocculant dosage, the maximum flocculation capacity was 2.29 g dye g^−1^ flocculant, which could transform into 0.24 mol dye: mol amine groups. In these conditions, the stoichiometric ratio of amine group to dye molecule was found to be 4 to 1. This result also indicates that charge neutralization is the primary mechanism of the dye flocculation, and thus, the flocculation mechanism and related floc composition can be determined (Figure 1c). Subsequently, we used these chitosan-containing textile sludges as precursors to prepare graphene-like carbon nanosheet (NSC-Fe-*x*) materials by simply pyrolyzing the mixture of sludge and Fe (III) salt and studied its application to electrocatalysis and energy storage.

### 3.2. Characterization of Samples

The XRD patterns of the NSC-Fe-*x* were displayed in Figure 2a. NSC-Fe-*x* shows diffraction peaks at 30.10° (220), 35.42° (311), 43.05° (400), and 62.52° (440), indicating the formation of Fe_3_O_4_ (JCPDS No. 19-0629) [22]. What is more, two diffraction peaks at 26.4° and 44° were observed, which are typical of the (002) and (100) planes of graphite carbon materials [23]. It is proverbial that the graphitization degree can be evaluated by the relative intensity of (002) peak, and a higher value indicates a higher degree of graphitization [24]. The highest stronger peak was at 26.4° of NSC-Fe-3, reflecting that the Fe (III) salt is crucial for converting chitosan-containing textile sludge to carbon materials with a high graphitization.

The graphitization degrees and NSC-Fe-*x* were studied by Raman spectroscopy (Figure 2b). The peaks at 1350, 1590, and 2901 cm^−1^ correspond to the D, G, and 2D bands, respectively, of the sample. The D band is ascribed to the degree of disorder in a sample, the G band is ascribed to graphitic sp^2^ hybridized carbon atoms, and the 2D band demonstrates the existence of a well-organized graphite structure in samples [25]. The graphitization degree can be evaluated by I_D_/I_G_, and a lower I_D_/I_G_ value indicates a higher degree of graphitization [26]. The intensity ratio values of I_D_/I_G_ for NSC-Fe-1, NSC-Fe-2, and NSC-Fe-3 were 0.90, 0.88, and 0.88, respectively. The lowest I_D_/I_G_ ratio (0.88) indicated the outstanding graphitic degree of NSC-Fe-2 and NSC-Fe-3, which was consistent with the XRD pattern. The results showed that the iron species mildly improved the degree of graphitization and were consistent with the results reported in other papers [27]. Furthermore, the higher graphitization degree could boost electroconductibility and catalytic activity, thereby improving the ORR performance [28].

Figure 2c shows the Brunauer–Emmett–Teller (BET) N_2_ adsorption–desorption isotherms for all samples. All samples exhibited type-IV characteristics with H4 hysteresis loops at 0.4 < P/P_0_ < 1.0, pointing the existence of micropores and mesopores. Detailed textural parameters are exhibited in Appendix A. The specific surface areas (SSAs) of the as-prepared NSC-Fe-2 (499.60 m^2^ g^−1^) were larger than that of NSC-Fe-1 and NSC-Fe-3. The low SSA of NSC-Fe-1 is due to the small amount of Fe (III) salt precursor forming insufficient Fe-related species. When increasing the amount of Fe (III) salt, the SSAs of NSC-Fe-3 is also lower to that of the NSC-Fe-2. It is indicated that an excess Fe (III) salt causes the particles of Fe species aggregated during calcination, thereby reducing the SSA. The results indicated that a proper amount of iron can increase the SSA of the materials, which was consistent with previous work [29]. It has been reported that high SSA materials can introduce more active sites to boost ORR activity and supercapacitor performance [30]. In addition, the pore size structure of the material is also a crucial parameter in determining the electrochemical properties of an electrode material [31]. The pore diameter was calculated by the Horvaih–Kawazoe (HK) and Barrett–Joyner–Halenda (BJH) methods [32], and the micropore and mesoporous pore sizes of NSC-Fe-2 were concentrated at 0.36 and 3.73 nm, respectively (Figure 2d and Appendix A). Previous research has shown that the hierarchical micropores/mesoporous pore size contributes to the effective transmission of ORR-related substances, because the mass transfer properties primarily depend on the porosity [33]. Hence, NSC-Fe-2 materials with high specific surface area (SSA) and a relatively concentrated pore structure will contribute to supercapacitors and ORR performance.

Figure 3a and Appendix A show the SEM images of iron-doped NSC-Fe-*x* samples with different mass. When the mass ratio of Fe (III) salt/sewage sludge is 0.1, NSC-Fe-1 (Appendix A) displays a rough and disordered sheet structure, which is related to the condensation of chitosan-containing textile sludge [34]. However, samples NSC-Fe-2 and NSC-Fe-3 (Appendix A), introduced with a high ratio of Fe (III) salt/sewage sludge, exhibited typical graphene-like nanosheet morphology. It may due to the addition of more iron species into the sewage sludge during the pyrolysis to enhance the degree of graphitization [35], which was in good agreement with the Raman and XRD. AFM analysis showed that the average thickness of NSC-Fe-2 flakes was ~3.4 nm (Figure 3b,c), corresponding to approximately nine graphitic layers. This further suggests that the graphene-like nanosheets were prepared successfully. However, this flocculant-assisted pyrolysis conversion method often requires large amounts of chitosan-based flocculants, which is a major drawback for graphene-like materials production on a commercial scale. The average thickness of as-prepared NSC-Fe-2 was thinner than that of NSC-Fe-1 (~22.8 nm, Appendix A) and NSC-Fe-3 (~5.3 nm, Appendix A), reflecting that Fe (III) salt can affect the thickness of the materials. Significantly, the thinner carbon nanosheet is more conducive to the catalytic activity for ORR [36]. So, the morphology of the NSC-Fe can be efficiently controlled by changing the mass ratio of Fe (III) salt/sewage sludge. The TEM images provide more details of the morphology and microstructure. In Figure 3d, we can see the thin carbon sheets structure of the NSC-Fe-2 sample, and the nanoparticles were uniformly dispersed in the thin carbon sheets. In addition, the graphene-like nanosheets structure contributes to the diffusion of the ions, thus improving the supercapacitors or the ORR activity [37]. The lattice spacing of the granules was measured to be 0.297 nm from the HRTEM images (Figure 3e and inset), corresponding to the (220) plane of Fe_3_O_4_; this is consistent with the XRD pattern. The element mapping of NSC-Fe-2 is shown in Figure 3f,k, further indicating that the Fe_3_O_4_ nanoparticles were decorated on carbon nanosheets. Previous studies have indicated that the Fe_3_O_4_ attached to carbon materials might enhance the stability of electrode materials for the ORR and supercapacitors [38]. Furthermore, the elemental mapping also indicated that N and S were uniformly distributed, which contributes to the electrocatalytic efficiency for the ORR and supercapacitors [39].

The XPS survey spectra in Figure 4a show the element composition of NSC-Fe-*x*. NSC-Fe-*x* shows a visible Fe 2p peak at ~710.65 eV, proving that Fe species were triumphantly doped into the carbon skeleton. The fitted Fe 2p spectra for the NSC-Fe-2 sample are exhibited in Figure 4b. The high-resolution Fe 2p spectral peaks at 709.6, 711.2, 712.7, 714.2, 718.2, 723.7, and 729.9 eV were consistent with Fe^0^ 2p_3/2_, Fe^2+^ 2p_3/2_, Fe^3+^ 2p_3/2_, satellite, Fe^0^ 2p_1/2_, Fe^2+^ 2p_1/2_, and Fe^3+^ 2p_1/2_, respectively. These peaks indicated that the Fe in NSC-Fe-2 was mainly present in the form of Fe_3_O_4_ [18]. The high-resolution C 1s peaks (Figure 4c) at 284.1, 285.0, 286.4 and 289.3 eV were consistent with C=C, C–N/C–O, C–S/C–O, and O–C=O/O–C–N, respectively. The strong C=C bond is associated with a good degree of graphitization in NSC-Fe-2, which is consistent with the measurements of the sharp G peak in the Raman spectrum (Figure 2b). In the N 1s spectra (Figure 4d), four types of N were exhibited: pyridinic-N (397.9 eV), pyrrolic-N (399.7 eV), graphitic-N (400.8 eV), and oxidized-N (405.1 eV) [40,41]. Previous researches have shown that pyridinic-N, pyrrolic-N, and graphitic-N can be used as effective active sites for the ORR [42,43]. Concretely, they can enhance diffusion-limited current density and onset potential [44,45]. The content of N species can be visualized according to Appendix A. It is obvious that NSC-Fe-2 possesses high percentages of pyrrolic-N (40.46%) and pyridinic-N (23.37%), as well as high graphitic-N (29.22%) content, which would be anticipated to enhance the overall electrochemical performance [46].

### 3.3. Oxygen Reduction Reaction (ORR) Activity

The ORR activity of the samples was examined in a 0.1 M KOH aqueous solution. First, CV was performed in O_2_-saturated and N_2_-saturated 0.1 M KOH at 50 mV s^−1^. As shown in Figure 5a, in contrast to the N_2_-saturated electrolyte, NSC-Fe-2 exhibited a reduction peak in the O_2_-saturated electrolyte, demonstrating that the NSC-Fe-2 possesses electrocatalytic activity for the ORR. Afterwards, linear sweep voltammetry (LSV) tests were performed with NSC-Fe-*x* and Pt/C at 1600 rpm and 5 mV s^−1^ (Figure 5b). As expected, NSC-Fe-2 had a higher onset potential (1.05 V) than NSC-Fe-1 and NSC-Fe-3, and its value is even higher than that of 20 wt.% Pt/C (0.98 V), and exceeds that of most ORR catalysts (Appendix A). It is notable, however, that our work focuses on electrode materials prepared from sludge flocs as precursors, which are essential for economic and social development because they can meet both environmental and sustainable energy development requirements. The reaction kinetics of NSC-Fe-2 were measured by an ORR polarization method at various rotation rates (Figure 5c). It was observed that the current density increased with the rotation speed (400–2025 rpm) because of more O_2_ flow at the electrode. The excellent performance of NSC-Fe-2 is derived from the formation of a heteroatom-doped carbon matrix by pyrolysis and the synergistic impact of various active sties. The electron transfer number (*n*) was analyzed using the Koutechy–Levich (K–L) equation:(4)$$\frac{1}{J}=\frac{1}{J_{L}}+\frac{1}{J_{K}}=\frac{1}{B}\omega^{-0.5}+\frac{1}{J_{K}}$$
(5)$$B=0.62nF{D_{0}}^{\frac{2}{3}}v^{\frac{-1}{6}}C_{0}$$
where *J*, *J_L_*, and *J_K_* are the tested current density, diffusion-limiting current density, and kinetic current density, respectively; *B* is the inverse of the slope of the K–L curve; *ω* is the angular speed of the electrode; the value of *F* is C mol^−1^ (Faraday constant); the value of *D*_0_ is 1.9 × 10^−5^ cm^2^ s^−1^ (O_2_ diffusion coefficient); the value of *ν* is 0.01 cm^2^ s^−1^ (kinetic viscosity); and the value of *C*_0_ is 1.2 × 10^−6^ mol L^−3^ (O_2_ bulk solubility).

The K–L plots for the NSC-Fe-2 (Figure 5d) observed *ω*^−1/2^ and *j*^−1^ had a linear relationship, and n was analyzed in the potential range of 0.3 to 0.6 V. The n values of the NSC-Fe-2 electrodes at different voltages were 3.9, 3.8, 3.8, and 3.9, evidencing nearly a 4e^−^ transfer. Durability is a significant controlling factor for fuel cells in actual applications. ESI Appendix A shows that the Tafel slope of NSC-Fe-2 is 91.4 mV dec^−1^, lower than that of Pt/C (92.8 mV dec^−1^). The lower Tafel slope also implies that the kinetic process of ORR is faster with NSC-Fe-2 than with Pt/C at high potentials. Hence, the durability of the NSC-Fe-2 and Pt/C catalysts were compared using i-t chronoamperometry. Compared with the Pt/C catalyst (68.2%), NSC-Fe-2 preserved 88.3% of its current density after 33,300 s (Figure 5e). The methanol crossover tolerance was also measured for NSC-Fe-2 and Pt/C (Figure 5f). After adding a 3 M methanol solution, the current of the Pt/C catalyst immediately decreased due to oxidization of methanol on the electrode surface. Notable, after adding the methanol solution, the current of the NSC-Fe-2 catalyst was still 98% of the initial current, showing good methanol crossover tolerance. As an ORR catalyst, the NSC-Fe-2 displayed a superior durability and methanol tolerance relative to the commercial 20 wt.% Pt/C catalyst. The outstanding ORR activity is mainly attributed to the higher graphitization degree, the thinner graphene-like structure and the higher SSA, which can expose a rich active sites and provide a more active place to transport electrolyte solutions [47].

### 3.4. Supercapacitor Performance

Figure 6a displays the CV curves of NSC-Fe-*x* at 50 mV s^−1^, and the enclosed area of the curve is proportional to the capacitance. Compared with the NSC-Fe-1, and NSC-Fe-3 electrodes, the NSC-Fe-2 electrode exhibits a much larger curve area, demonstrating that the NSC-Fe-2 electrode shows a superior specific capacitance. Figure 6b exhibits the CV curves of NSC-Fe-2 at various scan speeds. Impressively, the shape of the CV curves for all samples (Figure 6b and Appendix A) barely changed with increasing scanning speed (from 5 to 100 mV s^−1^), reflecting that all samples had superior electrochemical reversibility. Figure 6c shows the GCD curves of NSC-Fe-*x* at 1 A g^−1^. The long discharge time of NSC-Fe-2 indicates a superior specific capacity, which is in good agreement with the CV results. The NSC-Fe-2 showed outstanding capacitive behavior, and the specific capacity was 195 F g^−1^ at 1 A g^−1^, as can be seen from Figure 6d. The symmetrical GCD curves at various current densities clearly demonstrate the ideal capacitive characteristic of NSC-Fe-2 [48]. Figure 6e displays the capacitance retention of NSC-Fe-*x* at different current densities. Interestingly, NSC-Fe-2 still exhibited a good specific capacity of 132 F g^−1^ at 10 A g^−1^. The superior capacity of NSC-Fe-2 originates from its relatively concentrated porosity, which allows electrolyte diffusion, creates advantageous ion transmission channels and ensures ions can contact the electrode surface at various charge/discharge speeds. The Appendix A displays the Nyquist plots of the NSC-Fe-1, NSC-Fe-2, and NSC-Fe-3 electrodes. The Nyquist plots are composed of a portion of a semicircle in the high-frequency region that corresponds to the charge transfer resistance (R_ct_), and a straight line in the low-frequency region that is related to the capacitive behavior of the electrode. NSC-Fe-2 showed a smaller semicircle at high-frequencies than that of NSC-Fe-1 and NSC-Fe-3, which means that its charge transfer resistance was relatively lower than that of NSC-Fe-1 and NSC-Fe-3, which indicates low interfacial, electrolyte, and charge transfer resistances. To further illustrate the merits of using NSC-Fe-2 as an electrode material, we also performed GCD tests at 2 A g^−1^ to assess the stability of the NSC-Fe-2 electrode. After 2500 cycles, a high specific capacitance retention of 96.6% was achieved, revealing excellent cycling properties (Figure 6f). The high capacitance of NSC-Fe-2 electrode can be attributed to the following reasons. (1) The higher SSA and graphitization are favorable for charge transfer and enhance the conductivity [49]. (2) The presence of synergistic effects from codoping of iron and nitrogen, which enhanced the reactivity and wettability of the material [50]. (3) The thinner graphene-like structure is conducive to the adsorption and diffusion of ions [51].

## 4. Conclusions

In summary, a chitosan derivative containing tertiary amine functionality was synthesized and used as an effective flocculant in removal of anionic dye from aqueous solution. The composition and structure of textile sludge was investigated by stoichiometric charge neutralization between anionic dyes and cationic flocculant. Subsequently, graphene-like carbon nanosheets were synthesized through simple one-step carbonization of the chitosan-containing textile sludge and Fe (III) salt. The present electrode material exhibited a promising onset potential (1.05 V) for the ORR that was better than that of commercial 20 wt.% Pt/C (0.98 V). The NSC-Fe-2 catalyst followed a 4e^−^ pathway and displayed stability and methanol tolerance superior to those of Pt/C. At the same time, our electrode material also displayed an outstanding specific capacitance (195 F g^−1^ at 1 A g^−1^) and good cycling stability (loss of 3.4% after 2500 cycles). Considering the amount of sludge generated from textile wastewater treatment factories every day in China, this paper offers a viable method for converting textile sludge into available carbon materials with potential use in fuel cell, supercapacitor and photocatalysis applications.

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
