# Peer review of "Flocculant-Assisted Synthesis of Graphene-Like Carbon Nanosheets for Oxygen Reduction Reaction and Supercapacitor"

_nanomaterials, 2019, doi:10.3390/nano9081135_

Round 1
Reviewer 1 Report
The manuscript “Flocculant-Assisted Synthesis of Graphene…” by Zhang and coworkers that is submitted to Nanomaterials presents a new application for conversion of waste textile dying materials into value-added graphene for broader applications. The general methods for conversion of biomass to graphene-containing materials is well known. But the application to textile wastes appears to be novel and not previously published. The application is a simple extension of known prior work. One issue with the potential use of this method to conversion of textile wastes is that it requires 5x the amount of flocculant to be used relative to the waste to be treated. Thus, this is not an economically feasible approach to disposal/conversion of textile wastes. This should shortcoming must be acknowledged within the manuscript. The work is still worth publication. But there will need to be significant advances before this type of technology can ever be used on a production scale.
The characterization of the graphene material properties and performance presented in this paper is good and state-of-the-art. The English language is not good. The language is readable and communicates the results as needed. But there are numerous errors and misusages throughout the manuscript. These should be cleaned up before this work is published. These include correct identification of acronyms on first usage within the manuscript, grammar errors throughout, and non-standard language and terminology.
Author Response
Response to reviewers
Reviewer: 1
Comment #1:
The manuscript “Flocculant-Assisted Synthesis of Graphene…” by Zhang and coworkers that is submitted to Nanomaterials presents a new application for conversion of waste textile dying materials into value-added graphene for broader applications. The general methods for conversion of biomass to graphene-containing materials is well known. But the application to textile wastes appears to be novel and not previously published. The application is a simple extension of known prior work. One issue with the potential use of this method to conversion of textile wastes is that it requires 5x the amount of flocculant to be used relative to the waste to be treated. Thus, this is not an economically feasible approach to disposal/conversion of textile wastes. This should shortcoming must be acknowledged within the manuscript. The work is still worth publication. But there will need to be significant advances before this type of technology can ever be used on a production scale.
Response to comment #1:
We highly appreciate the valuable comments given by reviewer. Cost of the method used is still high, and we will consider reducing the processing cost of subsequent work in order to achieve industrialization.
Comment #2:
The characterization of the graphene material properties and performance presented in this paper is good and state-of-the-art. The English language is not good. The language is readable and communicates the results as needed. But there are numerous errors and misusages throughout the manuscript. These should be cleaned up before this work is published. These include correct identification of acronyms on first usage within the manuscript, grammar errors throughout, and non-standard language and terminology.
Response to comment #2:
We appreciate the reviewer’s suggestions. As suggested by the reviewer, we have tried our best to improve grammar errors throughout, the acronyms on first usage, and terminology. The alterations have been marked with highlighted red color in the manuscript.
Reviewer 2 Report
Authors report on the synthesis of graphene-like carbon nanosheets for oxygen reduction reaction and supercapacitor. The manuscript is well written and can be accepted after minor revision. It would be worth doing electrochemical impedance spectroscopy on these materials to find out equivalent circuits and other important parameters such as charge transfer resistance, series resistance, etc.
Author Response
Response to reviewers
Reviewer: 2
Comment #1:
Authors report on the synthesis of graphene-like carbon nanosheets for oxygen reduction reaction and supercapacitor. The manuscript is well written and can be accepted after minor revision. It would be worth doing electrochemical impedance spectroscopy on these materials to find out equivalent circuits and other important parameters such as charge transfer resistance, series resistance, etc.
Response to comment #1:
Thanks for your good suggestion. According to the referee’s suggestion, electrochemical impedance spectroscopy (EIS) spectra was included in the Supplementary Information (ESI Figure S8†) as follows:
Figure S8 Nyquist plot of the EIS for NSC-Fe-1, NSC-Fe-2 and NSC-Fe-3.
The corresponding discussion has been marked in red in the revised manuscript (Line 348-355) and revised supplementary information, and also given below. The ESI Figure S8† displays the Nyquist plots of the NSC-Fe-1, NSC-Fe-2 and NSC-Fe-3 electrodes. The Nyquist plots are composed of a portion of a semicircle in the high-frequency region that corresponds to the charge transfer resistance (Rct), and a straight line in the low frequency region that is related to the capacitive behavior of the electrode. NSC-Fe-2 showed a smaller semicircle at high-frequencies than that of NSC-Fe-1 and NSC-Fe-3, which means that its charge transfer resistance was relatively lower than that of NSC-Fe-1 and NSC-Fe-3, which indicates low interfacial, electrolyte and charge transfer resistances.

Reviewer 3 Report
This paper, dealing with the preparation of graphene-Like carbon nanosheets graphene-like carbon nanosheets for electrochemical application, is very interesting, and, a part some mistakes, well written. It was a pleasure to read it. Some minor remarks:
p. 1 line 25 (abstract) it is necessary to introducethe reference electrode for onset potential (vs RHE?)
The method used to determine the onset potential should be reported, since, in my opinion, the real onset potential of the best material is not 1.05, but lower than 1 V vs RHE
Bibliography should include some works of authoritative groups in ORR on Pt-free, as Atanassov (for XPS and the importance of some nitrogen groups), Zelenay and Mukerjee.
A suggestion: If authors calculate tafel slope and exchange currents, they will better support their conclusions
Author Response
Response to reviewers
Reviewer: 3
Comment #1:
This paper, dealing with the preparation of graphene-Like carbon nanosheets graphene-like carbon nanosheets for electrochemical application, is very interesting, and, a part some mistakes, well written. It was a pleasure to read it. Some minor remarks:
Response to comment #1:
We highly appreciate the valuable comments given by reviewer.
Comment #2:
p. 1 line 25 (abstract) it is necessary to introduce the reference electrode for onset potential (vs RHE?)
Response to comment #2:
Thanks for your good suggestion. As suggested by the reviewer, we have added the reference electrode for onset potential (0.98 V vs RHE) in Line 26 in revised manuscript.
Comment #3:
The method used to determine the onset potential should be reported, since, in my opinion, the real onset potential of the best material is not 1.05, but lower than 1 V vs RHE
Response to comment #3:
We appreciate the reviewer’s suggestions. The definitions of the onset potential vary from publication to publication. A potential value corresponding to 5% of the diffusion-limited current density (JL) has been proposed as one definition for the onset potential [1]. A potential at which the current density exceeds a threshold value of 0.1 mA cm-2 was also suggested [2]. Fe-N-C HNSs showed the onset potential at 1.046 V (>1 V) with a current density of 0.1 mA cm-2 [3]. Different onset potential definitions may lead to different results. It was very reasonable to obtain the onset potential at 1.05 V in this work at a current density of 0.1 mA cm-2. We added a potential with a current density of 0.1 mA cm-2 to the electrochemical test section as the onset potential, in Line 148-149 on Page 4 in revised manuscript.
References
1. Xia, W.; Mahmood, A.; Liang, Z.; Zou, R.; Guo, S. Earth-Abundant Nanomaterials for Oxygen Reduction. Angew. Chem. Int. Ed. 2016, 55, 2650-2676.
2. Daems, N.; Sheng, X.; Vankelecom, I.F.J.; Pescarmona, P.P. Metal-free doped carbon materials as electrocatalysts for the oxygen reduction reaction. J. Mater. Chem. A, 2014, 2, 4085-4110.
3. Chen, Y.; Li, Z.; Zhu, Y.; Sun, D.; Liu, X.; Xu, L.; Tang, Y. Atomic Fe Dispersed on N-Doped Carbon Hollow Nanospheres for High-Efficiency Electrocatalytic Oxygen Reduction. Adv. Mater. 2019, 31, e1806312.
Comment #4:
The Bibliography should include some works of authoritative groups in ORR on Pt-free, as Atanassov (for XPS and the importance of some nitrogen groups), Zelenay and Mukerjee.
Response to comment #4:
Thanks for your good suggestion. We have added some corresponding references for the Bibliography. The corresponding discussion has been marked in revised manuscript (Line 281-283) and also given below.
References
1. Artyushkova, K.; Serov, A.; Doan, H.; Danilovic, N.; Capuano, C.B.; Sakamoto, T.; Kishi, H.; Yamaguchi, S.; Mukerjee, S.; Atanassov, P. Application of X-ray photoelectron spectroscopy to studies of electrodes in fuel cells and electrolyzers. J. Electron Spectrosc. Relat. Phenom. 2019, 231, 127-139.
2. Hossen, M.M.; Artyushkova, K.; Atanassov, P.; Serov, A. Synthesis and characterization of high performing Fe-N-C catalyst for oxygen reduction reaction (ORR) in Alkaline Exchange Membrane Fuel Cells. J. Power Sources 2018, 375, 214-221.
3. Kim, D.; Zussblatt, N.P.; Chung, H.T.; Becwar, S.M.; Zelenay, P.; Chmelka, B.F. Highly Graphitic Mesoporous Fe,N-Doped Carbon Materials for Oxygen Reduction Electrochemical Catalysts. ACS Appl. Mater. Interfaces 2018, 10, 25337-25349.
4. Ferrandon, M.; Kropf, A.J.; Myers, D.J.; Artyushkova, K.; Kramm, U.; Bogdanoff, P.; Wu, G.; Johnston, C.M.; Zelenay, P. Multitechnique Characterization of a Polyaniline–Iron–Carbon Oxygen Reduction Catalyst. J. Phys. Chem. C 2012, 116, 16001-16013.
5. Artyushkova, K.; Serov, A.; Rojas-Carbonell, S.; Atanassov, P. Chemistry of Multitudinous Active Sites for Oxygen Reduction Reaction in Transition Metal–Nitrogen–Carbon Electrocatalysts. J. Phys. Chem. C 2015, 119, 25917-25928.
Comment #5:
A suggestion: If authors calculate tafel slope and exchange currents, they will better support their conclusions.
Response to comment #5:
Thanks for your good suggestion. According to the referee’s suggestion, tafel plots was included in the Supplementary Information (ESI Figure S6†) as follows:
Figure S6 Tafel plots of NSC-Fe-1, NSC-Fe-2, NSC-Fe-3 and commercial Pt/C catalysts.
The corresponding discussion has been marked in red in the revised manuscript (Line 318-321) and revised supplementary information, and also given below. ESI Figure S6† shows that the Tafel slope of NSC-Fe-2 is 91.4 mV dec-1, lower than that of Pt/C (92.8 mV dec-1). The lower Tafel slope also implies that the kinetic process of NSC-Fe-2 is faster than Pt/C at high potentials in ORR. In this work, we main focused on develop energy materials by environmental wastes and pay less attention to the excessive electrochemical performance of carbon materials. So, we did not calculate the exchange current of materials. We apologize for your question.
